# Cross-Episodic Curriculum for Transformer Agents

**Lucy Xiaoyang Shi**[1*], **Yunfan Jiang**[1*], **Jake Grigsby**[2], **Linxi "Jim" Fan**[3†], **Yuke Zhu**[2 3†]

[1]Stanford University    [2]The University of Texas at Austin    [3]NVIDIA Research

[*]Equal contribution   [†]Equal advising

## Abstract

We present a new algorithm, Cross-Episodic Curriculum (CEC), to boost the learning efficiency and generalization of Transformer agents. Central to CEC is the placement of *cross-episodic* experiences into a Transformer's context, which forms the basis of a curriculum. By sequentially structuring online learning trials and mixed-quality demonstrations, CEC constructs curricula that encapsulate learning progression and proficiency increase across episodes. Such synergy combined with the potent pattern recognition capabilities of Transformer models delivers a powerful *cross-episodic attention* mechanism. The effectiveness of CEC is demonstrated under two representative scenarios: one involving multi-task reinforcement learning with discrete control, such as in DeepMind Lab, where the curriculum captures the learning progression in both individual and progressively complex settings, and the other involving imitation learning with mixed-quality data for continuous control, as seen in RoboMimic, where the curriculum captures the improvement in demonstrators' expertise. In all instances, policies resulting from CEC exhibit superior performance and strong generalization. Code is open-sourced on the project website `cec-agent.github.io` to facilitate research on Transformer agent learning.

## 1 Introduction

The paradigm shift driven by foundation models [8] is revolutionizing the communities who study sequential decision-making problems [80], with innovations focusing on control [2, 45, 38, 9], planning [76, 32, 33, 78, 17], pre-trained visual representation [57, 50, 67, 51, 82], among others. Despite the progress, the data-hungry nature makes the application of Transformer [75] agents extremely challenging in data-scarce domains like robotics [52, 53, 19, 38, 9]. This leads us to the question: Can we maximize the utilization of limited data, regardless of their optimality and construction, to foster more efficient learning?

To this end, this paper introduces a novel algorithm named *Cross-Episodic Curriculum* (CEC), a method that explicitly harnesses the shifting distributions of multiple experiences when organized into a curriculum. The key insight is that sequential *cross-episodic* data manifest useful learning signals that do not easily appear in any separated training episodes.[1] As illustrated in Figure 1, CEC realizes this through two stages: 1) formulating curricular sequences to capture (a) the policy improvement on single environments, (b) the learning progress on a series of progressively harder environments, or (c) the increase of demonstrators' proficiency; and 2) causally distilling policy improvements into the model weights of Transformer agents through *cross-episodic attention*. When a policy is trained to predict actions at current time steps, it can trace back beyond ongoing trials and internalize improved behaviors encoded in curricular data, thereby achieving efficient learning

---

[1]Following the canonical definition in Sutton and Barto [73], we refer to the sequences of agent-environment interaction with clearly identified initial and terminal states as "episodes". We interchangeably use "episode", "trial", and "trajectory" in this work.

37th Conference on Neural Information Processing Systems (NeurIPS 2023).

Figure 1: **Cross-episodic curriculum for Transformer agents.** CEC involves two major steps: *1) Preparation of curricular data.* We order multiple experiences such that they explicitly capture curricular patterns. For instance, they can be policy improvement in single environments, learning progress in a series of progressively harder environments, or the increase of the demonstrator's expertise. *2) Model training with cross-episodic attention.* When training the model to predict actions, it can trace back beyond the current episode and internalize the policy refinement for more efficient learning. Here each $\tau$ represents an episode (trajectory). $\hat{a}$ refers to actions predicted by the model. Colored triangles denote causal Transformer models.

and robust deployment when probed with visual or dynamics perturbations. Contrary to prior works like Algorithm Distillation (AD, Laskin et al. [42]) which, at test time, samples and retains a single task configuration across episodes for in-context refinement, our method, CEC, prioritizes zero-shot generalization across a distribution of test configurations. With CEC, agents are evaluated on a new task configuration in each episode, emphasizing adaptability to diverse tasks.

We investigate the effectiveness of CEC in enhancing sample efficiency and generalization with two representative case studies. They are: 1) Reinforcement Learning (RL) on DeepMind Lab (DM-Lab) [5], a 3D simulation encompassing visually diverse worlds, complicated environment dynamics, ego-centric pixel inputs, and joystick control; and 2) Imitation Learning (IL) from mixed-quality human demonstrations on RoboMimic [53], a framework designed to study robotic manipulation with proprioceptive and external camera observations and continuous control. Despite RL episodes being characterized by state-action-reward tuples and IL trajectories by state-action pairs, our method exclusively employs state-action pairs in its approach.

In challenging embodied navigation tasks, despite significant generalization gaps (Table 1), our method surpasses concurrent and competitive method Agentic Transformer (AT, Liu and Abbeel [47]). It also significantly outperforms popular offline RL methods such as Decision Transformer (DT, Chen et al. [13]) and baselines trained on expert data, with the same amount of parameters, architecture, and data size. It even exceeds RL oracles directly trained on test task distributions by $50\%$ in a *zero-shot* manner. CEC also yields robust embodied policies that are up to $1.6\times$ better than RL oracles when zero-shot probed with unseen environment dynamics. When learning continuous robotic control, CEC successfully solves two simulated manipulation tasks, matching and outperforming previous well-established baselines [53, 25, 41]. Further ablation reveals that CEC with cross-episodic attention is a generally effective recipe for learning Transformer agents, especially in applications where sequential data exhibit moderate and smooth progression.

# 2 Cross-Episodic Curriculum: Formalism and Implementations

In this section, we establish the foundation for our cross-episodic curriculum method by first reviewing the preliminaries underlying our case studies, which encompass two representative scenarios in sequential decision-making. Subsequently, we formally introduce the assembly of curricular data and the specifics of model optimization utilizing cross-episodic attention. Lastly, we delve into the practical implementation of CEC in the context of these two scenarios.

## 2.1 Preliminaries

**Reinforcement learning.** We consider the setting where source agents learn through trial and error in partially observable environments. Denoting states $s \in \mathcal{S}$ and actions $a \in \mathcal{A}$, an agent interacts in a Partially Observable Markov Decision Process (POMDP) with the transition function $p(s_{t+1}|s_t, a_t) : \mathcal{S} \times \mathcal{A} \to \mathcal{S}$. It observes $o \in \mathcal{O}$ emitted from observation function $\Omega(o_t|s_t, a_{t-1}) : \mathcal{S} \times \mathcal{A} \to \mathcal{O}$ and receives scalar reward $r$ from $R(s, a) : \mathcal{S} \times \mathcal{A} \to \mathbb{R}$. Under the episodic task setting, RL seeks to learn a parameterized policy $\pi_\theta(\cdot|s)$ that maximizes the return over a fixed length $T$ of interaction steps: $\pi_\theta = \arg\max_{\theta \in \Theta} \sum_{t=0}^{T-1} \gamma^t r_t$, where $\gamma \in [0, 1)$ is a discount factor. Here we follow the canonical definition of an episode $\tau$ as a series of environment-agent interactions with length $T$, $\tau := (s_0, a_0, r_0, \ldots, s_{T-1}, a_{T-1}, r_{T-1}, s_T)$, where initial states $s_0$ are sampled from initial state distribution $s_0 \sim \rho_0(s)$ and terminal states $s_T$ are reached once the elapsed timestep exceeds $T$. Additionally, we view all RL tasks considered in this work as goal-reaching problems [39, 26] and constrain all episodes to terminate upon task completion. It is worth noting that similar to previous work [42], training data are collected by source RL agents during their online learning. Nevertheless, once the dataset is obtained, our method is trained *offline* in a purely supervised manner.

**Imitation learning.** We consider IL settings with existing trajectories composed only of state-action pairs. Furthermore, we relax the assumption on demonstration optimality and allow them to be crowdsourced [10, 12, 11]. Data collected by operators with varying expertise are therefore unavoidable. Formally, we assume the access to a dataset $\mathcal{D}^N := \{\tau_1, \ldots, \tau_N\}$ consisting of $N$ demonstrations, with each demonstrated trajectory $\tau_i := (s_0, a_0, \ldots, s_{T-1}, a_{T-1})$ naturally identified as an episode. The goal of IL, specifically of behavior cloning (BC), is to learn a policy $\pi_\theta$ that accurately models the distribution of behaviors. When viewed as goal-reaching problems, BC policies can be evaluated by measuring the success ratio in completing tasks [26].

## 2.2 Curricular Data Assembly and Model Optimization

Meaningful learning signals emerge when multiple trajectories are organized and examined cross-episodically along a curriculum axis. This valuable information, which is not easily discernible in individual training episodes, may encompass aspects such as the improvement of an RL agent's navigation policy or the generally effective manipulation skills exhibited by operators with diverse proficiency levels. With a powerful model architecture such as Transformer [75, 16], such emergent and valuable learning signals can be baked into policy weights, thereby boosting performance in embodied tasks.

For a given embodied task $\mathcal{M}$, we define its curriculum $\mathcal{C}_\mathcal{M}$ as a collection of trajectories $\tau$ consisting of state-action pairs. A series of ordered levels $[\mathcal{L}_1, \ldots, \mathcal{L}_L]$ partitions this collection such that $\bigcup_{l \in \{1,\ldots,L\}} \mathcal{L}_l = \mathcal{C}_\mathcal{M}$ and $\bigcap_{\forall i,j \in \{1,\ldots,L\}, i \neq j} \mathcal{L}_{\{i,j\}} = \emptyset$. More importantly, these ordered levels characterize a curriculum by encoding, for example, learning progress in single environments, learning progress in a series of progressively harder environments, or the increase of the demonstrator's expertise.

With a curriculum $\mathcal{C}_\mathcal{M} := \{\tau_i\}_{i=1}^N$ and its characteristics $[\mathcal{L}_1, \ldots, \mathcal{L}_L]$, we construct a curricular sequence $\mathcal{T}$ that spans multiple episodes and captures the essence of gradual improvement in the following way:

$$\mathcal{T} := \bigoplus_{l \in \{1,\ldots,L\}} \left[ \tau^{(1)}, \ldots, \tau^{(C)} \right], \quad \text{where} \quad C \sim \mathcal{U}\left(\llbracket |\mathcal{L}_l| \rrbracket\right) \quad \text{and} \quad \tau^{(c)} \sim \mathcal{L}_l. \tag{1}$$

The symbol $\oplus$ denotes the concatenation operation. $\mathcal{U}(\llbracket K \rrbracket)$ denotes a uniform distribution over the discrete set $\{k \in \mathbb{N}, k \leq K\}$. In practice, we use values smaller than $|\mathcal{L}_l|$ considering the memory consumption.

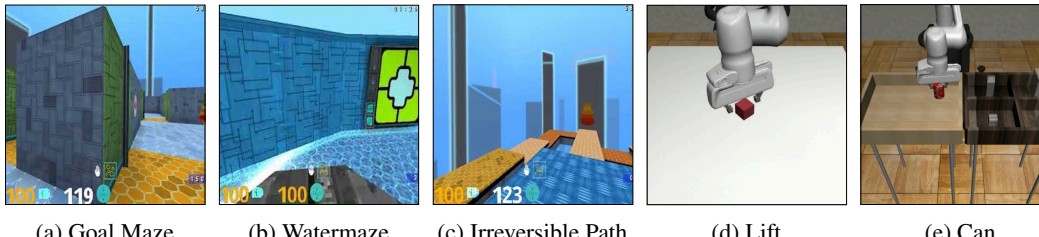

|(a) Goal Maze | (b) Watermaze | (c) Irreversible Path | (d) Lift | (e) Can |

Figure 2: We evaluate our method on five tasks that cover challenges such as exploration and planning over long horizons in RL settings, as well as object manipulation and continuous control in IL settings. Figures are from Beattie et al. [5] and Mandlekar et al. [53].

We subsequently learn a causal policy that only depends on cross-episodic historical observations $\pi_\theta(\cdot|o_{\leq t}^{(\leq n)})$. Note that this modeling strategy differs from previous work that views sequential decision-making as a big sequence-modeling problem [13, 37, 42, 38]. It instead resembles the causal policy in Baker et al. [4]. Nevertheless, we still follow the best practice [36, 60, 22] to provide previous action as an extra modality of observations in POMDP RL tasks.

We leverage the powerful attention mechanism of Transformer [75] to enable cross-episodic attention. Given observation series $O_t^{(n)} := \{o_0^{(1)}, \ldots, o_{\leq t}^{(\leq n)}\}$ (shorthanded as $O$ hereafter for brevity), Transformer projects it into query $Q = f_Q(O)$, key $K = f_K(O)$, and value $V = f_V(O)$ matrices, with each row being a $D$-dim vector. Attention operation is performed to aggregate information:

$$\text{Attention}(Q, K, V) = \text{softmax}(\frac{QK^\mathsf{T}}{\sqrt{D}})V. \tag{2}$$

Depending on whether the input arguments for $f_Q$ and $f_{\{K,V\}}$ are the same, attention operation can be further divided into self-attention and cross-attention. Since tasks considered in this work do not require additional conditioning for task specification, we follow previous work [4, 83] to utilize self-attention to process observation series. Nevertheless, ours can be naturally extended to handle, for example, natural language or multi-modal task prompts, following the cross-attention introduced in Jiang et al. [38].

Finally, this Transformer policy is trained by simply minimizing the negative log-likelihood objective $\mathcal{J}_{\text{NLL}}$ of labeled actions, conditioned on cross-episodic context:

$$\mathcal{J}_{\text{NLL}} = -\log \pi_\theta(\cdot|\mathcal{T}) = \frac{1}{|\mathcal{T}| \times T} \sum_{n=1}^{|\mathcal{T}|} \sum_{t=1}^{T} -\log \pi_\theta\left(a_t^{(n)}|o_{\leq t}^{(\leq n)}\right). \tag{3}$$

Regarding the specific memory architecture, we follow Baker et al. [4], Adaptive Agent Team et al. [1] to use Transformer-XL [16] as our model backbone. Thus, during deployment, we keep its hidden states propagating across test episodes to mimic the training settings.

## 2.3 Practical Implementations

We now discuss concrete instantiations of CEC for 1) RL with DMLab and 2) IL with RoboMimic. Detailed introductions to the benchmark and task selection are deferred to Sec. 3. We investigate the following three curricula, where the initial two pertain to RL, while the final one applies to IL:

**Learning-progress-based curriculum.** In the first instantiation, inspired by the literature on learning progress [54, 27, 65, 40], we view the progression of learning agents as a curriculum. Concretely, we train multi-task PPO agents [70, 63] on tasks drawn from test distributions. We record their online interactions during training, which faithfully reflect the learning progress. Finally, we form the *learning-progress-based curriculum* by sequentially concatenating episodes collected at different learning stages. Note that this procedure is different from Laskin et al. [42], where for each environment, the learning dynamics of *multiple* single-task RL agents has to be logged. In contrast, we only track a *single* multi-task agent per environment.

**Task-difficulty-based curriculum.** In the second instantiation, instead of taking snapshots of RL agents directly trained on test configurations, we collect learning progress on a series of easier

Table 1: **Generalization gaps between training and testing for DMLab levels.** Note that agents resulting from task-difficulty-based curricula are not trained on test configurations. Therefore, their performance should be considered as *zero-shot*.

| Level Name | Difficulty Parameter | Test Difficulty | Training Difficulty | | | | |
|---|---|---|---|---|---|---|---|
| | | | Ours (Learning Progress) | Ours (Task Difficulty) | BC w/ Expert Data | RL (Oracle) | Curriculum RL (Oracle) |
| Goal Maze | Room Numbers | 20 | 20 | 5→10→15 | 20 | 20 | 5→10→15→20 |
| Watermaze | Spawn Radius | 580 | 580 | 150→300→450 | 580 | 580 | 150→300→450→580 |
| Irreversible Path | Built-In Difficulty | .9 | .9 | .1→.3→.5→.7 | .9 | .9 | .1→.3→.5→.7→.9 |

but progressively harder tasks. For instance, in an embodied navigation task, the test configuration includes 20 rooms. Rather than logging source agents' learning progression in the 20-room maze, we record in a series of mazes with 5, 10, and 15 rooms. We then structure stored episodes first following learning progress and then the increase of layout complexity. This practice naturally creates a *task-difficulty-based curriculum*, which resembles curriculum RL that is based on task difficulty [54, 58]. We find it especially helpful for hard-exploration problems where the source RL agent does not make meaningful progress.

**Expertise-based curriculum.** For the setting of IL from mixed-quality demonstrations, we instantiate a curriculum based on demonstrators' expertise. This design choice is motivated by literature on learning from heterogeneous demonstrators [6, 81], with the intuition that there is little to learn from novices but a lot from experts. To realize this idea, we leverage the Multi-Human dataset from RoboMimic [53]. Since it contains demonstrations collected by human demonstrators with varying proficiency, we organize offline demonstration trajectories following the increase of expertise to construct the *expertise-based curriculum*.

## 3 Experimental Setup

In this section, we elaborate on the experimental setup of our case studies. Our investigation spans two representative and distinct settings: 1) online reinforcement learning with 3D maze environments of DMLab [5], and 2) imitation learning from mixed-quality human demonstrations of RoboMimic [53]. For each of them, we discuss task selection, baselines, and training and evaluation protocols. Teasers of these tasks are shown in Figure 2.

### 3.1 Task Settings and Environments

**DeepMind Lab** [5] is a 3D learning environment with diverse tasks. Agents spawn in visually complex worlds, receive ego-centric (thus partially observable) RGB pixel inputs, and execute joystick actions. We consider three levels from this benchmark: Goal Maze, Watermaze [56], and Sky Maze with Irreversible Path. They challenge agents to explore, memorize, and plan over a long horizon. Their goals are similar — to navigate in complicated mazes and find a randomly spawned goal, upon which sparse rewards will be released. Episodes start with randomly spawned agents and goals and terminate once goals are reached or elapsed steps have exceeded pre-defined horizons.

**RoboMimic** [53] is a framework designed for studying robot manipulation and learning from demonstrations. Agents control robot arms with fixed bases, receive proprioceptive measurements and image observations from mounted cameras, and operate with continuous control. We evaluate two simulated tasks: "Lift" and "Can". In the "Lift" task, robots are tasked with picking up a small cube. In the "Can" task, robots are required to pick up a soda can from a large bin and place it into a smaller target bin. Episodes start with randomly initialized object configuration and terminate upon successfully completing the task or exceeding pre-defined horizons.

### 3.2 Baselines

The primary goal of these case studies is to assess the effectiveness of our proposed cross-episodic curriculum in increasing the sample efficiency and boosting the generalization capability of Transformer agents. Therefore, in online RL settings, we compare against source RL agents which generate training data for our method and refer to them as *oracles*. These include a) PPO agents directly

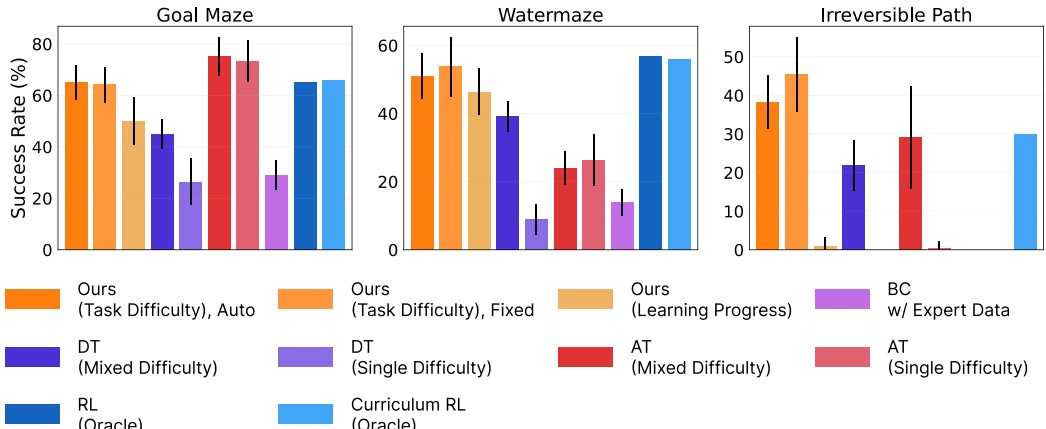

Figure 3: **Evaluation results on DMLab.** Our CEC agents perform comparable to RL oracles and on average outperform other baseline methods. On the hardest task Irreversible Path where the RL oracle and BC baseline completely fail, our agents outperform the curriculum RL oracle by $50\%$ even in a *zero-shot* manner. For our methods, DT, AT, and the BC w/ expert data baselines, we conduct 20 independent evaluation runs, each consisting of 100 episodes for Goal Maze and Watermaze and 50 episodes for Irreversible Path due to longer episode length. We test RL oracles for 100 episodes. The error bars represent the standard deviations over 20 runs.

trained on test task distributions, denoted as "**RL (Oracle)**" hereafter, and b) curriculum PPO agents that are gradually adapted from easier tasks to the test difficulty, which is referred to as "**Curriculum RL (Oracle)**". Furthermore, we compare against one concurrent and competitive method Agentic Transformer [47], denoted as "**AT**". It is closely related to our method, training Transformers on sequences of trajectory ascending sorted according to their rewards. We also compare against popular offline RL method Decision Transformer [13], denoted as "**DT**". Additionally, we include another behavior cloning agent that has the same model architecture as ours but is trained on optimal data without cross-episodic attention. This baseline is denoted as "**BC w/ Expert Data**". For the case study on IL from mixed-quality demonstrations, we adopt the most competing approach, **BC-RNN**, from Mandlekar et al. [53] as the main baseline. We also include comparisons against other offline RL methods [44] such as Batch-Constrained Q-learning (**BCQ**) [25] and Conservative Q-Learning (**CQL**) [41].

### 3.3 Training and Evaluation

We follow the best practice to train Transformer agents, including adopting AdamW optimizer [49], learning rate warm-up and cosine annealing [48], etc. Training is performed on NVIDIA V100 GPUs. During evaluation, for agents resulting from our method, each run involves several test rollouts to fill the context. We keep hidden states of Transformer-XL [16] propagating across episodes. We run other baselines and oracles for 100 episodes to estimate their performances. For our methods on RL settings, we compute the maximum success rate averaged across a sliding window over all test episodes to account for in-context improvement. The size of the sliding window equals one-quarter of the total test episodes. These values are averaged over 20 runs to constitute the final reporting metric. For our methods on the IL setting, since all training data are successful trajectories, we follow Mandlekar et al. [53] to report the maximum success rate achieved over the course of training, directly averaged over test episodes.

## 4 Experiments

We aim to answer the following four research questions through comprehensive experiments.

1. To what extent can our cross-episodic curriculum increase the sample efficiency of Transformer agents and boost their generalization capability?
2. Is CEC consistently effective and generally applicable across distinct learning settings?
3. What are the major components that contribute to the effectiveness of our method?

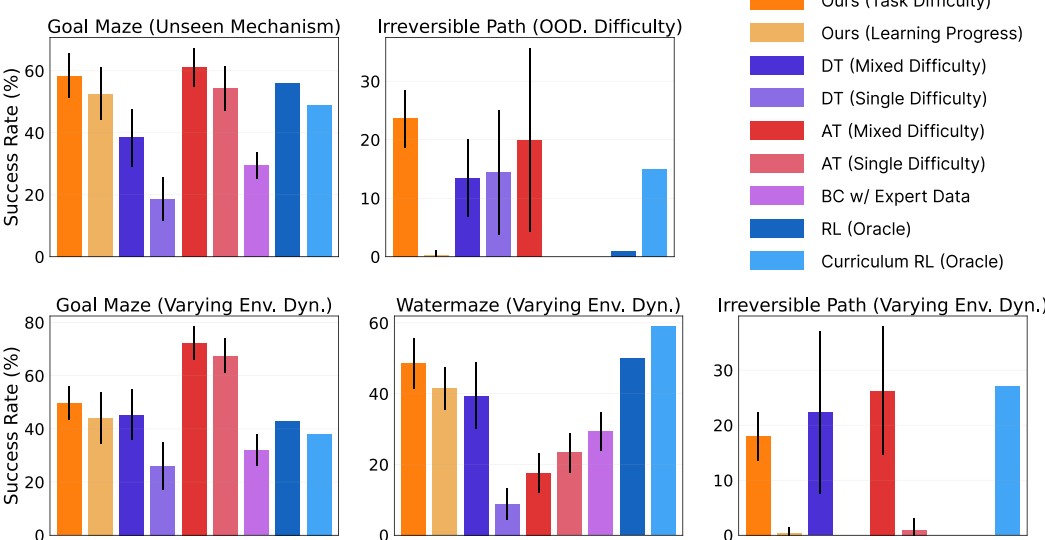

Figure 4: **Generalization results on DMLab.** *Top row*: Evaluation results on Goal Maze with unseen maze mechanism and Irreversible Path with out-of-distribution difficulty levels. *Bottom row*: Evaluation results on three levels with environment dynamics differing from training ones. CEC agents display robustness and generalization across various dimensions, outperforming curriculum RL oracles by up to $1.6\times$. We follow the same evaluation protocol as in Figure 3. The error bars represent the standard deviations over 20 runs.

## 4.1 Main Evaluations

We answer the first two questions above by comparing learned agents from our method against 1) Reinforcement Learning (RL) oracles in online RL settings and 2) well-established baselines on learning from mixed-quality demonstrations in the Imitation Learning (IL) setting.

We first examine agents learned from learning-progress-based and task-difficulty-based curricula in challenging 3D maze environments. The first type of agent is denoted as "**Ours (Learning Progress)**". For the second type, to ensure that the evaluation also contains a series of tasks with increasing difficulty, we adopt two mechanisms that control the task sequencing [58]: 1) fixed sequencing where agents try each level of difficulty for a fixed amount of times regardless of their performance and 2) dynamic sequencing where agents are automatically promoted to the next difficulty level if they consecutively succeed in the previous level for three times. We denote these two variants as "**Ours (Task Difficulty), Fixed**" and "**Ours (Task Difficulty), Auto**", respectively. Note that because the task-difficulty-based curriculum does not contain any training data on test configurations, these two settings are zero-shot evaluated on test task distributions. We summarize these differences in Table 1. We denote AT and DT trained on data consisting of a mixture of task difficulties as "**AT (Mixed Difficulty)**" and "**DT (Mixed Difficulty)**". Note that these data are the same used to train "Ours (Task Difficulty)". Similarly, we denote AT and DT directly trained on test difficulty as "AT (Single Difficulty)" and "DT (Single Difficulty)". These data are the same used to train "Ours (Learning Progress)".

**Cross-episodic curriculum results in sample-efficient agents.**  As shown in Figure 3, on two out of three examined DMLab levels, CEC agents perform comparable to RL oracles and outperform the BC baselines trained on expert data by at most $2.8\times$. On the hardest level Irreversible Path where agents have to plan the route ahead and cannot backtrack, both the BC baseline and RL oracle fail. However, our agents succeed in proposing correct paths that lead to goals and significantly outperform the curriculum RL oracle by $50\%$ even in a *zero-shot* manner. Because CEC only requires environment interactions generated during the course of training of online source agents (the task-difficulty-based curriculum even contains fewer samples compared to the curriculum RL, as illustrated in Table 1), the comparable and even better performance demonstrates that our method yields highly sample-efficient embodied policies. On average, our method with task-difficulty-based curriculum performs the best during evaluation (Table A.5), confirming the benefit over the

Table 2: **Evaluation results on RoboMimic.** Visuomotor policies trained with our expertise-based curriculum outperform the most competing history-dependent behavior cloning baseline, as well as other offline RL algorithms. For our method on the Lift task, we conduct 5 independent runs each with 10 rollout episodes. On the Can task, we conduct 10 independent runs each with 5 rollout episodes due to the longer horizon required to complete the task. Standard deviations are included.

| Task | Ours | BC-RNN [53] | BCQ [25] | CQL [41] |
|------|------|-------------|----------|----------|
| Lift | $100.0 \pm 0.0$ | $100.0 \pm 0.0$ | $93.3 \pm 0.9$ | $11.3 \pm 9.3$ |
| Can | $100.0 \pm 0.0$ | $96.0 \pm 1.6$ | $77.3 \pm 6.8$ | $0.0 \pm 0.0$ |

Table 3: **Ablation on the importance of cross-episodic attention.** Transformer agents trained with the same curricular data but without cross-episodic attention degrade significantly during evaluation, suggesting its indispensable role in learning highly performant policies.

| | DMLab | | | RoboMimic | |
|---|---|---|---|---|---|
| | Goal Maze | Watermaze | Irreversible Path | Lift | Can |
| Ours | $65.2 \pm 6.7$ | $50.9 \pm 6.6$ | $38.2 \pm 7.0$ | $100.0 \pm 0.0$ | $100.0 \pm 0.0$ |
| Ours w/o Cross-Episodic Attention | $35.0 \pm 7.1$ | $20.0 \pm 2.5$ | $3.8 \pm 4.9$ | $75.9 \pm 12.3$ | $99.3 \pm 0.9$ |

concurrent AT approach that leverages chain-of-hindsight experiences. When compared to DT, it outperforms by a significant margin, which suggests that our cross-episodic curriculum helps to squeeze learning signals that are useful for downstream decision-making.

**Cross-episodic curriculum boosts the generalization capability.** To further investigate whether CEC can improve generalization at test time, we construct settings with unseen maze mechanisms (randomly open/closed doors), out-of-distribution difficulty, and different environment dynamics. See the Appendix, Sec. C.2 for the exact setups. As demonstrated in Figure 4, CEC generally improves Transformer agents in learning robust policies that can generalize to perturbations across various axes. On three settings where the BC w/ Expert Data baseline still manages to make progress, CEC agents are up to $2\times$ better. Compared to oracle curriculum RL agents, our policies significantly outperform them under three out of five examined scenarios. It is notable that on Irreversible Path with out-of-distribution difficulty, CEC agent is $1.6\times$ better than the curriculum RL oracle trained on the same data. These results highlight the benefit of learning with curricular contexts. On average, our method surpasses the concurrent AT baseline and achieves significantly better performance than other baselines (Table A.6). This empirically suggests that CEC helps to learn policies that are robust to environmental perturbations and can quickly generalize to new changes.

**Cross-episodic curriculum is effective across a wide variety of learning scenarios.** We now move beyond RL settings and study the effectiveness of the expertise-based curriculum in the IL setting with mixed-quality demonstrations. This is a common scenario, especially in robotics, where demonstrations are collected by human operators with varying proficiency [52]. As presented in Table 2, visuomotor policies trained with the expertise-based curriculum are able to match and outperform the well-established baseline [53] on two simulated robotic manipulation tasks and achieve significantly better performance than agents learned from prevalent offline RL algorithms [25, 41]. These results suggest that our cross-episodic curriculum is effective and broadly applicable across various problem settings. More importantly, it provides a promising approach to utilizing limited but sub-optimal data in data-scarce regimes such as robot learning.

## 4.2 Ablation Studies

In this section, we seek to answer the third research question to identify the components critical to the effectiveness of our approach. We focus on three parts: the importance of cross-episodic attention, the influence of curriculum granularity, and the effect of varying context length. Finally, we delve into the fourth question, identifying scenarios where CEC is expected to be helpful.

**Importance of cross-episodic attention.** The underlying hypothesis behind our method is that cross-episodic attention enables Transformer agents to distill policy improvement when mixed-optimality trajectories are viewed collectively. To test this, on DMLab levels and RoboMimic tasks, we train the same Transformer agents with the same curricular data and training epochs but

without cross-episodic attention. We denote such agents as "**Ours w/o Cross-Episodic Attention**" in Table 3. Results demonstrate that the ablated variants experience dramatic performance degradation on four out of five examined tasks, which suggests that naively behaviorally cloning sub-optimal data can be problematic and detrimental. Cross-episodic attention views curricular data collectively, facilitating the extraction of knowledge and patterns crucial for refining decision-making, thereby optimizing the use of sub-optimal data.

**Curriculum granularity.** We perform this ablation with the task-difficulty-based curriculum on DMLab levels, due to the ease of adjusting granularity. We treat the curricula listed in the column "Ours (Task Difficulty)" in Table 1 as "Fine", and gradually make them coarser to study the impact. Note that we ensure the same amount of training data. See the Appendix, Sec. C.4 for how we define granularity levels "Medium" and "Coarse". We visualize the performance relative to the most fine-grained in Figure 5. The monotonic degradation of policy performance with respect to curriculum coarseness suggests that fine-grained curricula are critical for Transformer agents to mostly benefit from cross-episodic training.

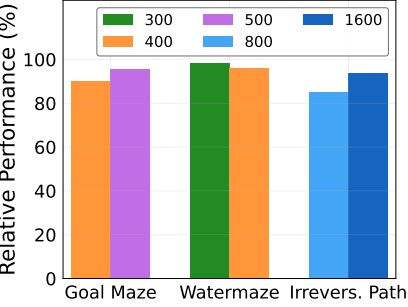

Figure 5: We compare the performance relative to agents trained with the fine-grained curricula. Performance monotonically degrades as task-difficulty-based curricula become coarser.

**Varying context length.** Lastly, we study the effect of varying context length on DMLab and visualize it in Figure 6. We normalize all performance values relative to those of "Ours (Task Difficulty), Auto" reported in Figure 3. It turns out that both too short and unnecessarily long context windows are harmful. On two out of three levels, using a shorter context decreases the performance even more. This finding coincides with Laskin et al. [42] that a sufficiently long Transformer context is necessary to retain cross-episodic information. Furthermore, we also discover that an unnecessarily long context is also harmful. We hypothesize that this is due to the consequent training and optimization instability.

**Curriculum selection based on task complexities and data sources.** For RL tasks, we recommend starting with the learning-progress-based curriculum. However, if the task itself is too challenging, such that source algorithms barely make progress, we recommend the task-difficulty-based curriculum. In IL settings, we further investigate the performance of the learning-progress-based curriculum on RoboMimic tasks considered in this work. Detailed setup and results are included in Appendix, Sec C.5. To summarize, if human demonstrations are available, even if they are generated to be heterogeneous in quality, we recommend using the expertise-based curriculum. However, in the absence of human demonstrations and only with access to machine-generated data (e.g., generated by RL agents), our learning-progress-based curriculum is recommended because it achieves non-trivial performance and significantly outperforms offline RL methods such as CQL [41].

Figure 6: Both short and unnecessarily long context windows decrease the performance. Numbers in the legend denote context lengths. Performance values are relative to those of "Ours (Task Difficulty), Auto" reported in Figure 3. "Irrevers. Path" stands for the task "Irreversible Path".

## 5 Related Work

**Sequential decision-making with Transformer agents.** There are many ongoing efforts to replicate the strong emergent properties demonstrated by Transformer models for sequential decision-making problems [80]. Decision Transformer [13] and Trajectory Transformer [37] pioneered this thread by casting offline RL [44] as sequence modeling problems. Gato [68] learns a massively multi-task agent that can be prompted to complete embodied tasks. MineDojo [22] and VPT [4] utilize numerous YouTube videos for large-scale pre-training in the video game *Minecraft*. VIMA [38] and RT-1 [9] build Transformer agents trained at scale for robotic manipulation tasks. BeT [71] and C-BeT [14] design novel techniques to learn from demonstrations with multiple modes with Trans-

formers. Our causal policy most resembles to VPT [4]. But we focus on designing learning techniques that are generally effective across a wide spectrum of learning scenarios and application domains.

**Cross-episodic learning.** Cross-episodic learning is a less-explored terrain despite that it has been discussed together with meta-RL [77] for a long time. $RL^2$ [18] uses recurrent neural networks for online meta-RL by optimizing multi-episodic value functions. Meta-Q-learning [21] instead learns multi-episodic value functions in an offline manner. Algorithm Distillation (AD) [42] and Adaptive Agent (AdA) [1] are two recent, inspiring methods in cross-episodic learning. Though at first glance our learning-progress-based curriculum appears similar to AD, significant differences emerge. Unlike AD, which focuses on in-context improvements at test time and requires numerous single-task source agents for data generation, our approach improves data efficiency for Transformer agents by structuring data in curricula, requiring only a single multi-task agent and allowing for diverse task instances during evaluations. Meanwhile, AdA, although using cross-episodic attention with a Transformer backbone, is rooted in online RL within a proprietary environment. In contrast, we focus on offline behavior cloning in accessible, open-source environments, also extending to IL scenarios unexplored by other meta-learning techniques. Complementary to this, another recent study [43] provides theoretical insight into cross-episodic learning.

**Curriculum learning.** Curriculum learning represents training strategies that organize learning samples in meaningful orders to facilitate learning [7]. It has been proven effective in numerous works that adaptively select simpler task [58, 74, 69, 62, 15, 55, 59, 46] or auxiliary rewards[35, 72]. Tasks are also parameterized to form curricula by manipulating goals [24, 30, 66], environment layouts[79, 3, 64], and reward functions [28, 34]. Inspired by this paradigm, our work harnesses the improving nature of sequential experiences to boost learning efficiency and generalization for embodied tasks.

# 6 Conclusion

In this work, we introduce a new learning algorithm named *Cross-Episodic Curriculum* to enhance the sample efficiency of policy learning and generalization capability of Transformer agents. It leverages the shifting distributions of past learning experiences or human demonstrations when they are viewed as curricula. Combined with cross-episodic attention, CEC yields embodied policies that attain high performance and robust generalization across distinct and representative RL and IL settings. CEC represents a solid step toward sample-efficient policy learning and is promising for data-scarce problems and real-world domains.

**Limitations and future work.** The CEC algorithm relies on the accurate formulation of curricular sequences that capture the improving nature of multiple experiences. However, defining these sequences accurately can be challenging, especially when dealing with complex environments or tasks. Incorrect or suboptimal formulations of these sequences could negatively impact the algorithm's effectiveness and the overall learning efficiency of the agents. A thorough exploration regarding the attainability of curricular data is elaborated upon in Appendix, Sec D.

In subsequent research, the applicability of CEC to real-world tasks, especially where task difficulty remains ambiguous, merits investigation. A deeper assessment of a demonstrator's proficiency trajectory — from initial unfamiliarity to the establishment of muscle memory — could offer a valuable learning signal. Moreover, integrating real-time human feedback to dynamically adjust the curriculum poses an intriguing challenge, potentially enabling CEC to efficiently operate in extended contexts, multi-agent environments, and tangible real-world tasks.

## Acknowledgments and Disclosure of Funding

We thank Guanzhi Wang and Annie Xie for helpful discussions. We are grateful to Yifeng Zhu, Zhenyu Jiang, Soroush Nasiriany, Huihan Liu, and Rutav Shah for constructive feedback on an early draft of this paper. We also thank the anonymous reviewers for offering us insightful suggestions and kind encouragement during the review period. This work was partially supported by research funds from Salesforce and JP Morgan.

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

# A  Model Architecture

In this section, we provide comprehensive details about the Transformer model architectures considered in this work. We implement all models in PyTorch [61] and adapt the implementation of Transformer-XL from VPT [4].

## A.1  Observation Encoding

Experiments conducted on both DMLab and RoboMimic include RGB image observations. For models trained on DMLab, we use a ConvNet [29] similar to the one used in Espeholt et al. [20]. For models trained on RoboMimic, we follow Mandlekar et al. [53] to use a ResNet-18 network [29] followed by a spatial-softmax layer [23]. We use independent and separate encoders for images taken from the wrist camera and frontal camera. Detailed model parameters are listed in Table A.1.

Table A.1: Model hyperparameters for vision encoders.

| Hyperparameter | Value |
|---|---|
| **DMLab** | |
| Image Size | $72 \times 96$ |
| Number of ConvNet Blocks | 1 |
| Channels per Block | [16, 32, 32] |
| Output Size | 256 |
| **RoboMimic** | |
| Image Size | $84 \times 84$ |
| Random Crop Height | 76 |
| Random Crop Width | 76 |
| Number of Randomly Cropped Patches | 1 |
| ConvNet Backbone | ResNet-18 [29] |
| Output Size | 64 |
| Spatial-Softmax Number of Keypoints | 32 |
| Spatial-Softmax Temperature | 1.0 |
| Output Size | 64 |

Since DMLab is highly partially observable, we follow previous work [20, 22, 4] to supply the model with previous action input. We learn 16-dim embedding vectors for all discrete actions.

To encode proprioceptive measurement in RoboMimic, we follow Mandlekar et al. [53] to not apply any learned encoding. Instead, these types of observation are concatenated with image features and passed altogether to the following layers. Note that we do not provide previous action inputs in RoboMimic, since we find doing so would incur significant overfitting.

## A.2  Transformer Backbone

We use Transformer-XL [16] as our model backbone, adapted from Baker et al. [4]. Transformer-XL splits long sequences into shorter sub-sequences that reduce the computational cost of attention while allowing the hidden states to be carried across the entire input by attending to previous keys and values. This feature is critical for the long sequence inputs necessary for cross-episodic attention. Detailed model parameters are listed in Table A.2.

## A.3  Action Decoding

To decode joystick actions in DMLab tasks, we learn a 3-layer MLP whose output directly parameterizes a categorical distribution. This action head has a hidden dimension of 128 with ReLU activations. The "Goal Maze" and "Irreversible Path" tasks have an action dimension of 7, while "Watermaze" has 15 actions. To decode continuous actions in RoboMimic, we learn a 2-layer MLP that parameterizes a Gaussian Mixture Model (GMM) with 5 modes that generates a 7-dimensional action. This network

Table A.2: Model hyperparameters for Transformer-XL.

| Hyperparameter | Value (DMLab) | Value (RoboMimic) |
|---|---|---|
| Hidden Size | 256 | 400 |
| Number of Layers | 4 | 2 |
| Number of Heads | 8 | 8 |
| Pointwise Ratio | 4 | 4 |

has a hidden dimension of 400 with ReLU activations. During deployment, we employ the "low-noise evaluation" trick [31].

# B  Training Details and Hyperparameters

All experiments are conducted on cluster nodes with NVIDIA V100 GPUs. We utilize DDP (distributed data parallel) to accelerate the training if necessary. Training hyperparameters are listed in Table A.3.

Table A.3: Hyperparameters used during training.

| Hyperparameter | Value (DMLab) | Value (RoboMimic) |
|---|---|---|
| Learning Rate | 0.0005 | 0.0001 |
| Warmup Steps | 1000 | 0 |
| LR Cosine Annealing Steps | 100000 | N/A |
| Weight Decay | 0.0 | 0.0 |

# C  Experiment Details

## C.1  DMLab Main Experiment

Our DMLab main experiment is conducted on three levels with task IDs

- `explore_goal_locations_large`,
- `rooms_watermaze`,
- and `skymaze_irreversible_path_hard`.

We use no action repeats during training and evaluation. For experiments with varying task difficulty, we select difficulty parameters "room numbers", "spawn radius", and "built-in difficulty" for these three levels, respectively. We adopt environment wrappers and helper functions from Petrenko et al. [63] to flexibly and precisely maneuver task difficulties.

Due to different task horizons, we tune the context length of Transformer-XL models and vary curricular trajectories accordingly. These differences are summarized in Table A.4.

RL oracles serve as source agents used to generate training data for our methods and the "BC w/ Expert Data" baseline. They are trained with the PPO [70] implementation from Petrenko et al. [63]. The "BC w/ Expert Data" baselines have the same model architecture, training hyperparameters, and amount of training data as our method, but are trained solely on trajectories generated by the best performing RL oracles without cross-episodic attention.

## C.2  DMLab Generalization

This series of experiments probe the zero-shot generalization capabilities of embodied agents in unseen maze configurations, out-of-distribution difficulty levels, and varying environment dynamics. For the task "Goal Maze w/ Unseen Mechanism", we use the level with task ID

Table A.4: Experiment details on DMLab tasks. Columns "Epoch" denote the exact training epochs with best validation performance. We select these checkpoints for evaluation. For task-difficulty-based curriculum, the column "Training Trajectories" with $n \times m$ entries means $n$ trajectories per difficulty level ($m$ levels in total). The column "Sampled Episodes" with $[i, j]$ entries means we first determine the number of episodes per difficulty level by uniformly sampling an integer from $[i, j]$ (inclusively).

| Level | Context | Task-Difficulty-Based Curriculum | | | Learning-Progress-Based Curriculum | | |
|---|---|---|---|---|---|---|---|
| Name | Length | Epoch | Training Trajectories | Sampled Episodes | Epoch | Training Trajectories | Sampled Episodes |
| Goal Maze | 500 | 84 | 100 x 3 | [1, 5] | 88 | 300 | 9 |
| Watermaze | 400 | 89 | 100 x 3 | [1, 5] | 80 | 300 | 9 |
| Irreversible Path | 1600 | 90 | 100 x 4 | [1, 3] | 97 | 400 | 8 |

Table A.5: Evaluation results on DMLab, averaged over three tasks (Figure 3).

| Ours (Task Difficulty), Auto | Ours (Task Difficulty), Fixed | Ours (Learning Progress) | DT (Mixed Difficulty) | DT (Single Difficulty) | AT (Mixed Difficulty) | AT (Single Difficulty) | BC w/ Expert Data | RL (Oracle) | Curriculum RL (Oracle) |
|---|---|---|---|---|---|---|---|---|---|
| 51.4 | **54.4** | 32.4 | 35.3 | 11.7 | 42.7 | 33.4 | 14.2 | 40.6 | 50.6 |

Table A.6: Generalization results on DMLab, averaged over five settings (Figure 4).

| Ours (Task Difficulty) | Ours (Learning Progress) | DT (Mixed Difficulty) | DT (Single Difficulty) | AT (Mixed Difficulty) | AT (Single Difficulty) | BC w/ Expert Data | RL (Oracle) | Curriculum RL (Oracle) |
|---|---|---|---|---|---|---|---|---|
| **39.6** | 27.8 | 31.8 | 13.6 | 39.4 | 29.2 | 18.1 | 30.0 | 37.6 |

`explore_obstructed_goals_large`, which adds randomly opened and closed doors into the maze while ensuring a valid path to the goal always exists. An example of an agent's ego-centric observation is visualized in Figure A.1.

The task "Irreversible Path (OOD. Difficulty)" corresponds to configurations with the built-in difficulty of 1 (agents are only trained on difficulty up to 0.9, as noted in Table 1). For tasks with varying environment dynamics, we directly test agents with an action repeat of 2. This is different from the training setting with no action repeat.

## C.3 RoboMimic Main Experiment

We leverage the Multi-Human (MH) dataset from Mandlekar et al. [53]. It consists of demonstrations collected by operators with varying proficiency. We construct the expertise-based curriculum by following the order of "worse operators, okay operators, then better operators". We use a context length of 200 for both tasks. There are 90 trajectories per expertise level. To determine the number of trajectories per expertise level when constructing curricular data, we uniformly sample an integer from $[1, 5]$ (inclusively). The "Lift" and "Can" tasks are solved after training for 33 epochs and 179 epochs, respectively. We control for the same number of training epochs in subsequent ablation studies.

## C.4 Ablation Study on Curriculum Granularity

We perform this ablation with the task-difficulty-based curriculum on DMLab levels due to the ease of adjusting granularity. The definition of varying levels of curriculum coarseness is listed in Table A.7.

Table A.7: Definitions of varying levels of curriculum coarseness.

| Level Name | Difficulty Parameter | Test Difficulty | Fine | Medium | Coarse |
|---|---|---|---|---|---|
| Goal Maze | Room Numbers | 20 | 5→10→15 | 5→10 | 5→15 |
| Watermaze | Spawn Radius | 580 | 150→300→450 | 150→300 | 150→450 |
| Irreversible Path | Built-In Difficulty | 0.9 | .1→.3→.5→.7 | .1→.5→.7 | .1→.3→.5 |

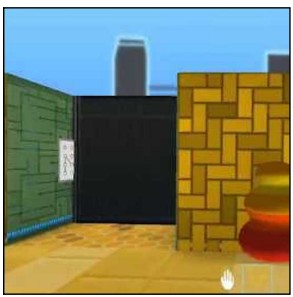

Figure A.1: **A visualization of the task "Goal Maze (Unseen Mechanism)".** It includes doors that are randomly opened or closed.

Table A.8: Results show the performance of different curricula on two robotic manipulation tasks: Lift and Can. Standard deviations are included.

| Task | Expertise-Based Curriculum | Learning-Progress-Based Curriculum | CQL [41] |
|------|---------------------------|-----------------------------------|----------|
| Lift | $100.0 \pm 0.0$ | $32.0 \pm 17.0$ | $2.7 \pm 0.9$ |
| Can | $100.0 \pm 0.0$ | $30.0 \pm 2.8$ | $0.0 \pm 0.0$ |
| Average | **100.0** | 31.0 | 1.4 |

### C.5  Comparison of Curricula in RoboMimic

In IL settings, we further explored the efficacy of various curricula. For the RoboMimic tasks examined, we employed a learning-progress-based curriculum, ensuring the total training trajectories matched those of the expertise-based curriculum (i.e., 270 trajectories per task). All other parameters remained consistent, with the training data derived from RoboMimic's machine-generated dataset.

Table A.8 indicates that when heterogeneous-quality human demonstrations are accessible, the expertise-based curriculum is preferable due to its superior performance over the learning-progress-based approach. Conversely, without expert demonstrations and relying solely on machine-generated data, the learning-progress-based curriculum is still commendable. It offers noteworthy results and surpasses offline RL methods like CQL [41], even though CQL is trained on the full RoboMimic dataset, encompassing 1500 trajectories for the Lift task and 3900 for the Can task.

## D  Feasibility of Obtaining Curricular Data

The challenge of accurately orchestrating a curriculum is non-trivial and hinges on various factors. In the present work, three curriculum designs are introduced and validated, each with its practical considerations and underlying assumptions, discussed herein.

**Learning-Progress-Based Curriculum.**  RL agents typically exhibit monotonic improvement over training epochs, thereby naturally producing incrementally better data. The curriculum here is devised through a series of checkpoints throughout the training duration, necessitating no supplementary assumptions for its formulation.

**Task-Difficulty-Based Curriculum.**  In contexts where environmental difficulty is parameterizable, curricula can be structured through a schedule, determined by the relevant difficulty parameter, as demonstrated within this work. In scenarios lacking parameterized difficulty, alternatives such as methods proposed by Kanitscheider et al. [40] may be employed. The application of our method to tasks where difficulty is not explicitly characterized presents an intriguing avenue for future research.

**Expertise-Based Curriculum.**  A notable limitation resides in the requisite to estimate demonstrators' proficiency. While some IL benchmarks, e.g., RoboMimic [53], come pre-equipped with proficiency labels, a broader application of our method necessitates an approximation of proficiency. One plausible approach entails ranking trajectories via completion time. Furthermore,

a demonstrator's proficiency is likely to organically improve—from initial unfamiliarity with teleoperation systems or tasks, to a stage of executing data collection with muscle memory [52]. This progression potentially provides a rich learning signal conducive for CEC application.

# E   Broader Impact

Our Cross-Episodic Curriculum can significantly enhance Transformer agent learning but carries potential societal impacts. The efficiency of our method depends on the curriculum's design. If the curriculum unintentionally reflects biases, it could lead to the amplification of these biases in learned policies, potentially perpetuating unfair or discriminatory outcomes in AI-driven decisions. Furthermore, the computational intensity of our approach at evaluation could contribute to increased energy usage, which has implications for the environmental footprint of AI applications.

