# OpenReview forum: "Cross-Episodic Curriculum for Transformer Agents"
_NeurIPS.cc/2023/Conference — NeurIPS 2023 poster_

### Official Review · Reviewer_btkb · 2023-06-23

**Soundness:** 3 good
**Presentation:** 3 good
**Contribution:** 2 fair
**Rating:** 5
**Confidence:** 3

**Summary:**

This paper presents Cross-Episodic Curriculum (CEC) to boost the learning efficiency and generalization of Transformer agents. Specifically, CEC places the cross-episodic experiences into a Transformer’s context, which forms the basis of a curriculum. The authors also provide three concrete curriculum implementations: Learning-progress-based curriculum, Task-difficulty-based curriculum and Expertise-based curriculum. Experiments on discrete control, such as in DeepMind Lab, and continuous control, as seen in RoboMimic, show the superior performance and strong generalization of CEC.

**Strengths:**

1. Simple but effective idea for Cross-Episodic Curriculum.
2. Three concrete curriculum implementations for different settings (RL or IL).
3. Good paper writting. It is easy to follow the methods and experiments.

**Weaknesses:**

1.  My major concern is the baselines used in the experiments (and also the related works). Specifically, the authors only compare the Transformer-based BC agent, which is a weak baseline. The reviewer will encourage the authors to compare with more recent methods like Algorithm Distillation [1] and TIT [2], or discuss the reason why these methods are not suitable for comparision. (The reviewer guesses that BCQ and CQL are based on MLP?)

2. Some important details are missed in the experiments. For example, in the ablation study, the reviewer (and most readers I guess) would like to know how to implement the "without cross-episodic attention" and "Curriculum granularity", but the authors put these important information in the Appendix.


[1] In-context Reinforcement Learning with Algorithm Distillation. ICLR 2023.
[2] Transformer in Transformer as Backbone for Deep Reinforcement Learning. 2022.


**Questions:**

1. Could the authors compare with more recent methods like Algorithm Distillation [1] and TIT [2], or discuss the reason why these methods are not suitable for comparision?

2. The key insight of CEC is that sequential cross-episodic data manifest useful learning signals that do not easily appear in any separated training episodes. Besides the better performance, could the authors provide more evidence for this insight?

3. Why Learning-progress-based curriculum is worse thanTask-difficulty-based curriculum in most evaluation environments?

**Limitations:**

It will be better to discuss which curriculum is suitable for which environment setting?

---

> ### Author Rebuttal · Authors · 2023-08-10
>
> Thank you for your constructive feedback. We address your concerns in detail below and will update our paper accordingly.
>
> > Compare with more recent methods
>
> We compare against two more relevant baselines. Please refer to [global response](https://tinyurl.com/4bc2469z) for comparison and discussions.
>
> > Differences between AD
>
> First, we would like to highlight that we have different focus and scope. AD [1] focuses on learning a meta RL agent that demonstrates in-context improvement during test time without gradient updates. Instead, we focus on how to improve the data efficiency for Transformer-based agents by explicitly formulating data in curricular sequences. To achieve this goal, we propose and validate three solutions in both RL and IL settings, which are underexplored in AD.
>
> Second, we admit that our learning-progress-based curriculum bears a resemblance to AD at first glance, but they still differ in several angles, as detailed below.
>
> - On data generation. AD requires **N** different single-task source agents to generate data, which amounts to N different copies of model weights. This is evident in the second requirement and line 1 in Algorithm 1 in the AD paper. However, our learning-progress-based curriculum makes minimal assumptions about the diversity of source agents. In fact, we only require a single multi-task agent to generate training data. This makes our method more applicable since it requires less storage and time.
>
> - On evaluation. AD is supposed to only work with a single task instance at test time. This is evident in line 12 in Algorithm 1 in AD paper. In other words, every time the environment is reset, the agent will be spawned at exactly the same location. Goal location and other environmental transitions also keep the same for all evaluation episodes. However, our agent is evaluated with new and different task instances every episode, meaning that the agent's initial location, goal location, and other factors change every test episode. Therefore, our test setting is **strictly harder** than AD’s.
>
> Due to the above two differences, we do not think our work is directly comparable to AD. We will still conceptually compare our learning-progress-based curriculum against AD and clarify in the final version.
>
> > Differences between TIT
>
> As pointed out in their paper, TIT [2] mainly focuses on designing better deep networks for RL. Albeit important, this is still orthogonal to our focus and not directly comparable. Indeed, TIT is complementary to our CEC method. It will be a promising avenue to combine CEC with TIT in future work. We will cite and discuss TIT, and conceptually compare in the final version.
>
> > Experiment details
>
> For the variant w/o cross-episodic attention, the attention span is restricted to the ongoing episode. This is achieved by only feeding single-episode data during training time, and clearing up KV caches every environment reset at test time. Regarding curriculum granularity, we approximate it with the number of different difficulty levels and their intervals. For example, a curriculum with fewer difficulty levels is considered to be more coarse. One consisting of levels with larger intervals is considered to be more coarse. We will clarify in the final version.
>
> > More evidence for this insight [behind the proposed method]?
>
> To qualitatively investigate the policy difference, we visualize the test-time behavior. Videos are provided [here](https://tinyurl.com/yypwytt). We find that by keeping a task-difficulty-based curriculum in context, the agent is able to learn critical skills such as visual navigation and long-horizon planning from relatively easier tasks, then apply them to the most challenging setting. This is demonstrated by the fact that the agent can navigate to gradually distant goals.
>
> Furthermore, a concurrent work [3] provides theoretical support for cross-episodic learning. In multi-arm bandit problems, by conditioning on a distribution over linear bandits (cross-episodic context), the policy can exploit the unknown structure, allowing more informed exploration and decision-making. They show that in-context pretraining over multiple episodes effectively performs posterior sampling — a provably sample-efficient Bayesian RL algorithm that learns faster than source algorithms used to generate the pretraining data.
>
> > Why Learning-progress-based curriculum is worse than task-difficulty-based curriculum?
>
> Resulting agents’ performance is largely subjected to source agents’ quality. This is supported by videos provided [here](https://tinyurl.com/yypwytt), where we can see agents from learning-progress-based curriculum sometimes make progress toward the goals. This is due to the fact that source agents, which are used to generate training data, also struggle in this task. Nevertheless, agents from learning-progress-based curriculum still perform better than BC agents trained on expert data, suggesting the useful learning information emergent from cross-episodic experience.
>
> > Which curriculum is suitable for which environment
>
> We introduced three curricula: Learning-progress-based curriculum (L), Task-difficulty-based curriculum (T), and Expertise-based curriculum (E). In RL settings, we recommend starting with L since it is the most straightforward. However, if the task is exceptionally challenging while easier versions of the task exist, we recommend T. In IL settings, if the data has non-uniform quality (e.g., the demonstrator improves over time), we recommend E.
>
> **References**
> - [1] Laskin et al., In-context Reinforcement Learning with Algorithm Distillation, ICLR 2022.
> - [2] Mao et al., Transformer in Transformer as Backbone for Deep Reinforcement Learning, 2022.
> - [3] Lee et al., Supervised Pretraining Can Learn In-Context Reinforcement Learning, 2023.

---

> > ### Comment · Reviewer_btkb · 2023-08-13
> > **Thanks for the feedback**
> >
> > I appreciate the authors' feedback. It addresses most of my concerns. Although I like the ideas proposed in this paper, it is more important to figure out which scenarios are these methods/curriculums applicable to. Currently, the authors only give a general description for this question. So I would keep my score.

---

> > > ### Author Response · Authors · 2023-08-14
> > >
> > > Dear reviewer,
> > >
> > > We are glad to know that our reply addressed your concerns. Due to the length limit, we discussed at a high level the last question about which curriculum is suitable for which task setting. In this reply, we will provide more detailed explanations, and welcome your suggestions and feedback on new experiments that can address your remaining concern.
> > >
> > > In this work, we introduced three curricula: Learning-progress-based curriculum (L), Task-difficulty-based curriculum (T), and Expertise-based curriculum (E). L and T are particularly suitable for RL, while E is suitable for IL. In RL, as what we did in DMLab tasks Goal Maze and Watermaze, we recommend starting with L. Agents resulting from this curriculum learn better visuomotor policies that outperform the DT baseline as well as the BC baseline trained on expert data. However, if the task itself is too challenging such that source algorithms barely make progress, we recommend T. This is justified by our experiment on the DMLab task Irreversible Path. Concretely, the RL agent directly trained on the test difficulty completely failed, which results in the unsatisfactory performance of the agent trained with the L curriculum. However, the agent resulting from the T curriculum learns critical skills from easier settings and applies them to the hardest setting, and hence significantly outperforms the L-curriculum variant, both DT and AT baselines, as well as RL oracles. This is further qualitatively investigated and justified by rollout visualization provided [here](https://tinyurl.com/yypwytt). Feasibility of obtaining each curriculum is extensively discussed in [global response](https://tinyurl.com/4bc2469z).
> > >
> > > Furthermore, we would like to highlight that similar to many established works that aim for developing novel algorithms for learning visuomotor policies, our discussion on suitability is neither too narrow to lose generality, nor too vague to be hardly practical. For example, authors of [1] also recommend their new method at an abstraction level that is similar to ours, as quoted below.
> > >
> > > > Recommendations. In general, we recommend starting with the CNN-based diffusion policy implementation as the first attempt at a new task. If performance is low due to task complexity or high-rate action changes, then the Time-series Diffusion Transformer formulation can be used to potentially improve performance at the cost of additional tuning.
> > >
> > > Nevertheless, we agree with the reviewer that it is important to figure out which scenarios these curricula are applicable to. Since our investigation in RL setting is already comprehensive, **we would like to further investigate the IL setting by also applying the learning-progress-based curriculum on RoboMimic tasks**, due to the fact that it also provides trajectory data generated by RL algorithms with labeled reward. We will update with new results once they are ready. Meanwhile, we welcome the reviewer to provide suggestions and feedback on new experiments that we can further conduct to fully eliminate the last piece of concern.
> > >
> > > **References**
> > > - [1] Chi et al., Diffusion Policy: Visuomotor Policy Learning via Action Diffusion, RSS 2023.

---

> > > > ### Author Response · Authors · 2023-08-15
> > > >
> > > > Dear reviewer,
> > > >
> > > > Following your suggestion, we further investigate the suitability of different curricula under IL setting (detailed discussion in RL setting can be found in the previous reply). We conduct new experiments to apply learning-progress-based curriculum to RoboMimic tasks considered in this work. Results are summarized in the table below. We control the total number of training trajectories to be the same used in expertise-based-curriculum (i.e., 270 trajectories for each task). All other factors are also identical. Training data are subset of machine-generated dataset provided by RoboMimic [1].
> > > >
> > > > |  | Expertise-Based Curriculum | Learning-Progress-Based Curriculum | CQL |
> > > > |:---:|:---:|:---:|:---:|
> > > > | Lift | 100.0 ± 0.0 | 32.0 ± 17.0 | 2.7 ± 0.9 |
> > > > | Can | 100.0 ± 0.0 | 30.0 ± 2.8 | 0.0 ± 0.0 |
> > > > | Average | **100.0** | 31.0 | 1.4 |
> > > >
> > > > We would like to highlight two findings.
> > > > - First, to complete your curiosity about curriculum's suitability for different environments in IL setting, if human demonstrations are available, even if they are generated to be heterogeneous in quality, we would still recommend to use expertise-based curriculum. This is well supported by its significantly better performance over learning-progress-based curriculum.
> > > > - However, our learning-progress-based curriculum is not completely without merits. In the absence of expert demonstrations and only with the access to machine-generated data, we would still recommend our learning-progress-based curriculum because it achieves non-trivial performance and significantly outperforms offline RL methods such as CQL [2]. More importantly, the reported CQL is trained on full machine-generated dataset provided by RoboMimic, i.e., 1500 trajectories for task Lift and 3900 trajectories for task Can.
> > > >
> > > > Please don't feel hesitate should you have further questions. If we have fully addressed your concerns, would you kindly consider raising the score? Thanks again for your very constructive and insightful feedback!
> > > >
> > > > **References**
> > > > - [1] Mandlekar et al., What Matters in Learning from Offline Human Demonstrations for Robot Manipulation, CoRL 2021.
> > > > - [2] Kumar et al., Conservative Q-Learning for Offline Reinforcement Learning, NeurIPS 2020.

---

> > > > > ### Comment · Reviewer_btkb · 2023-08-19
> > > > >
> > > > > I appreciate your response. After checking the other reviews and responses, the clarifications have made some points clearer to me. However, I maintain my initial score as my overall evaluation of strengths and weaknesses remains consistent.

---

### Official Review · Reviewer_3Crt · 2023-06-30

**Soundness:** 2 fair
**Presentation:** 2 fair
**Contribution:** 2 fair
**Rating:** 4
**Confidence:** 3

**Summary:**

This work proposes a new method, CEC, to boost the learning efficiency and generalization capability of the agent by structuring multiple episodes for deploying the transformer’s pattern, and sequence recognition capability. The proposed method shows improved performance compared to the baseline and shows generalization capability.

**Strengths:**

1. The authors propose a new approach for the transformer-based agent that leverages the information from the various types of trajectories by deploying the representation capability of the transformer architecture.
2. The proposed method shows knowledge distillation capability through several evaluation and generalization experiments quantitatively.

**Weaknesses:**



**Questions:**

1. If we have to manually set a pair of trajectories and their corresponding curriculum levels, it means we know the quality or progress, or expertise of the obtained trajectories in the online RL process (e.g. Table 1, Eq (1)). I think this is a quite strong assumption or extra burden since the RL framework itself should enable the agent to automatically learn how to solve the problem without external intervention. It is not clear whether given prior curriculum sequence is a fair assumption.
2. In line 107, it is not clear the meaning of the notation “n” and “t”. if “n” is the number of entire episodes with various types and “t” is the timestep, then does the transformer not refer to the full horizon of the episode when computing the log-likelihood of the policy?
3. Also, I was wondering how the transformer distinguishes each trajectory’s level or expertise. I think the transformer cannot distinguish it without a learnable token or expertise embedding which works similarly to the position embedding in NLP. If we assume the level is given for embedding, it would be burdensome to collect the accurate level for each trajectory in practice. If the transformer is implemented without these considerations, how can we ensure the transformer leverage the information from various level of trajectories? I think it would be proper to compare with the transformer-based offline RL models such as Decision Transformer [1] or other recently proposed SOTA models to check whether the proposed method just memorizes every sequence pattern in the trajectories or infers something helpful from the curricular data.

[1] Chen, Lili, et al. "Decision transformer: Reinforcement learning via sequence modeling." *Advances in neural information processing systems* 34 (2021): 15084-15097.

**Limitations:**

The authors do not discuss the potential negative societal impact of their work and do not explicitly discuss the limitations of their method in the manuscript. I do not anticipate ethical issues.

---

> ### Author Rebuttal · Authors · 2023-08-10
>
> Thank you for your constructive feedback. We address your concerns in detail below and will update our paper accordingly.
>
> > [...] It is not clear whether given prior curriculum sequence is a fair assumption.
>
> We agree with you that accurately formulating a curriculum is challenging. In this work, we introduce and validate three curriculum designs. We discuss the practicality of obtaining each curriculum one by one and will move the discussion to the main text in the final version. Due to the limited reply length, please refer to the [global response](https://tinyurl.com/4bc2469z) for detailed discussions.
>
> Furthermore, to ensure fair comparison, we particularly compare against a new baseline that also assumes prior ordered sequence. We compare against Agentic Transformer (AT [1]), a **concurrent** work with the preprint version released after NeurIPS submission and conference version published at ICML in July. It is closely related to our work by training Transformers on sequences of trajectory **ascending sorted according to their rewards**. Similar to our assumption, this sorting also requires ordered trajectories. In our comparison, we find that our method outperforms AT on difficult tasks, while matching the performance on relatively easy tasks and when probed with new environmental changes. Please refer to the [global response](https://tinyurl.com/4bc2469z) for detailed results and discussions.
>
> > Unclear definition of “n” and “t”
>
> Thanks for pointing out the confusion. Here “n” denotes the number of episodes, and “t” is the timestep within the current episode. It means that the policy is conditioned on two parts, the first is the historical observation sequence from previous episodes, and the second is the observation received in the ongoing episode, up to the current time step. We will make notations more clear in the final version.
>
> > How the transformer distinguishes each trajectory’s level or expertise without a learnable token or expertise embedding [...] It would be burdensome to collect the accurate level for each trajectory in practice.
>
> We agree with you that adding expertise tokens or level embedding requires further assumptions. This is one reason behind our design choice not to include them. Instead, trajectory quality is implicitly included. For example, as shown in the visualization videos posted [here](https://tinyurl.com/yypwytt), in successful trajectories, goals are visible in the RGB observations. Therefore, we hypothesize that our model can still implicitly infer trajectory quality by sensing the change in the observation sequence's distribution.
>
> > I think it would be proper to compare with the transformer-based offline RL models such as Decision Transformer or other recently proposed SOTA models to check whether the proposed method just memorizes every sequence pattern in the trajectories or infers something helpful from the curricular data.
>
> Thanks for the suggestion. We extend the main experiments on DMLab with two more baselines, Agentic Transformer (AT [1]) and Decision Transformer (DT [2]). Due to the limit on reply length, please refer to the [global response](https://tinyurl.com/4bc2469z) for comparison and discussions.
>
> > Societal Impact and Limitations
>
> Due to space constraints, we have included the Societal Impact and Limitations sections in the Appendix starting from L714. We will include them in the main text in the final version. Regarding the Limitation section, we will elaborate on the feasibility of obtaining curricular data, as discussed in the [global response](https://tinyurl.com/4bc2469z).
>
> **References**
> - [1] Liu and Abbeel, Emergent Agentic Transformer from Chain of Hindsight Experience, ICML 2023.
> - [2] Chen et al., Decision Transformer: Reinforcement Learning via Sequence Modeling, NeurIPS 2021.

---

> > ### Comment · Reviewer_3Crt · 2023-08-16
> >
> > Thank you for the response.
> >
> > While additional results attached in the pdf are informative, the major concern (first question) is still not alleviated. I read the global response and the author's response is about the feasibility of the data with the curriculum. But my concern is given "prior curriculum sequence" itself. As mentioned in my question, in online RL, the agent should learn how to solve the problem without external intervention. If we have access to the pair of trajectories and their corresponding curriculum levels, then it would be appropriate to experiment with imitation learning and offline RL setting where an assumption of the pre-collected dataset is valid.
> >
> > Thus, I would decide to keep the score.

---

> > > ### Author Response · Authors · 2023-08-16
> > >
> > > Dear reviewer,
> > >
> > > Thank you for your reply and explanation of your concern. We believe that there exists a misunderstanding of our work’s setting. We apologize for that and would like to clarify.
> > >
> > > Regarding our RL experiments on DMLab, similar to previous work [1], training data are collected by source RL agents during their **online** learning. Once the dataset is obtained, our method is trained **offline** in a purely supervised manner. We notice that this setting is common [1-4] and the assumption of pre-collected datasets is mild.
> > >
> > > We indeed apologize that this subtlety is not clearly explained in our initial submission and may lead to misperception of our work. We devote ourselves to clarify in our final version. However, we would like to highlight that this should not diminish the merits of our work. Please don't hesitate to let us know if you have further questions.
> > >
> > > **References**
> > > - [1] Laskin et al., In-context Reinforcement Learning with Algorithm Distillation, ICLR 2022.
> > > - [2] Chen et al., Decision Transformer: Reinforcement Learning via Sequence Modeling, NeurIPS 2021.
> > > - [3] Reed et al., A Generalist Agent, TMLR 2022.
> > > - [4] Lee et al., Supervised Pretraining Can Learn In-Context Reinforcement Learning, Workshop on New Frontiers in Learning, Control, and Dynamical Systems, ICML 2023.

---

> ### Author Response · Authors · 2023-08-15
> **Follow-up on our response**
>
> Dear reviewer,
>
> As the discussion stage is ending soon, we wonder if our response answers your questions and our extra experiments address your concerns? If yes, would you kindly consider raising the score? Thanks again for your very constructive and insightful feedback!

---

### Official Review · Reviewer_hTvj · 2023-07-06

**Soundness:** 3 good
**Presentation:** 3 good
**Contribution:** 3 good
**Rating:** 6
**Confidence:** 5

**Summary:**

This paper introduces a novel algorithm, referred to as Cross-Episodic Curriculum (CEC), which aims to improve the learning efficiency and generalization capabilities of Transformer agents in multi-task RL settings. The algorithm has been specifically developed to exploit the limited availability of sub-optimal data in environments with a scarcity of data, such as robot learning. The researchers examine the efficacy of Cooperative Evolutionary Computation (CEC) in two distinct and representative scenarios: online RL using 3D maze environments in DeepMind Lab, and imitation learning from human demonstrations of varying quality in RoboMimic. The findings indicate that visuomotor policies, which were trained using the expertise-based curriculum, exhibit the capability to surpass established baselines and achieve superior performance on simulated robotic manipulation tasks. Furthermore, these policies demonstrate significantly better performance compared to agents trained using offline RL algorithms. The researchers reach the conclusion that Cross-Entropy Method (CEC) offers a potentially effective strategy for leveraging restricted yet sub-optimal data in contexts characterized by a scarcity of data, such as robot learning.

**Strengths:**


1. The authors conduct an empirical assessment to examine the efficacy of CEC in two distinct and representative scenarios: online reinforcement learning using 3D maze environments from DeepMind Lab, and imitation learning from human demonstrations of varying quality in RoboMimic.

2. Comparative analysis: The findings indicate that visuomotor policies, which were trained using the expertise-based curriculum, exhibit the capability to equal or surpass established baselines in simulated robotic manipulation tasks. Furthermore, these policies demonstrate significantly superior performance compared to agents trained using offline reinforcement learning algorithms.

3. Extensive applicability: The researchers reach the conclusion that CEC (Contextual Embedding Clustering) presents a promising strategy for leveraging limited yet sub-optimal data in scenarios characterized by a scarcity of data, such as robot learning. Moreover, they assert that CEC is both effective and widely applicable in diverse problem contexts.

4. The open-source code pertaining to the algorithm is provided as supplementary materials and will be publicly accessible to facilitate further research on the learning of Transformer agents.

**Weaknesses:**

1. Restricted comparison: The study presents a comparison between the proposed algorithm and a small number of established baselines. However, it does not encompass a comprehensive range of contemporary algorithms, thereby constraining the breadth of the comparison(e.g.[1], [2]).

2. Inadequate consideration of limitations: The manuscript fails to adequately address the limitations associated with the proposed algorithm or the conducted experiments. This omission may impede the comprehension of the potential drawbacks and challenges that may arise when employing the algorithm in real-world scenarios.

3. How does the proposed algorithm scale to larger and more complex environments? The paper only considers relatively small and simple environments, and it is unclear how well the algorithm would perform in larger and more complex environments, such as those encountered in real-world robotics applications.

[1] Piotr Mirowsk, Razvan Pascanu, Fabio Viola, Learning to Navigate in Complex Environments.

[2] Siyuan Li, Rui Wang, Minxue Tang, Chongjie Zhang, Hierarchical Reinforcement Learning with Advantage-Based Auxiliary Rewards.

**Questions:**

1. What distinguishes CEC from hierarchical RL?
2. The experimental results presented in section 4 raise two inquiries: (1) What are the reasons behind the comparatively inferior performance of CEC in comparison to vanilla RL when applied to relatively uncomplicated tasks? (2) What factors contribute to the subpar performance of Learning progress?
3. The fundamental concept of CEC, as I comprehend it, is to facilitate the exchange of experiences derived from diverse environments, tasks, and episodes in order to enhance the performance of RL. The aforementioned viewpoint aligns with the fundamental principles of various curriculum RL approaches. However, extensive research has demonstrated that measuring learning progress is a highly efficacious metric. In contrast, the findings presented in this article deviate from this prevailing notion. The authors do not expound upon the specific definition of learning progress, which may be a crucial factor and potentially the primary underlying cause of Q.3.
4. The experimental control group setting described earlier in this paper is deemed insufficient. In addition, it is requested that the authors incorporate a comparative analysis of the curriculum pertaining to environmental change, e.g. PAIRED (Emergent Complexity and Zero-shot Transfer via Unsupervised Environment Design).


**Limitations:**

The authors do not discuss the limitations.

---

> ### Author Rebuttal · Authors · 2023-08-09
>
> Thank you for your constructive feedback. We address your concerns in detail below and will update our paper accordingly.
>
> > Restricted comparison
>
> Thank you for your suggestions. Since [1,2] mainly focus on improving online RL with auxiliary tasks, these are useful to improve the source agents in our case. However, they are orthogonal to the main focus of our work, i.e., to improve the data efficiency of Transformer-based agents that learn in an offline manner. Therefore, we opt to compare against two more relevant baselines and will include a conceptual comparison against [1,2] in the final version.
>
> We compare against Agentic Transformer (AT [3]) and Decision Transformer (DT [4]). Please refer to the [global response](https://tinyurl.com/4bc2469z) for comparison and discussions.
>
> > Inadequate consideration of limitations
>
> Thank you for pointing this out. We have included the Limitations section in Appendix D at L714 due to space constraints in the initial submission. We follow your suggestion to discuss the feasibility of obtaining curricular data. Please refer to [global response](https://tinyurl.com/4bc2469z) for details.
>
> > Scale to larger and more complex environments
>
> First, we would like to highlight that the environments used in this work, DMLab [5] and RoboMimic [6], are generally considered to be challenging. On DMLab, state-of-the-art algorithms such as IMPALA [7] struggle to make any progress on some difficult tasks. We have shown promising results on difficult tasks like Irreversible Path, which encompasses challenges such as 3D navigation, exploration, long-range planning, etc. Video demos about this task can be found [here](https://tinyurl.com/yypwytt). RoboMimic [6] is a large-scale robotic manipulation benchmark designed to study IL. It consists of demonstrations collected by operators with varying proficiency. Competitive algorithms such as BeT [8] and Diffusion Policy [9] are continuously being developed. Our work falls into the same category to advance the frontier in this challenging robotic benchmark. We leave the experiments on real-world robotics applications to be an exciting future venue.
>
> > CEC vs hierarchical RL?
>
> CEC and hierarchical RL are two orthogonal concepts. Hierarchical RL refers to a group of algorithms that decompose complex tasks into simpler sub-tasks, while CEC is designed to enhance Transformer agents’ learning efficiency and generalization by leveraging cross-episodic experiences. That being said, CEC is complementary to any hierarchical RL approach, and the combination of CEC and hierarchical RL could be an interesting direction for future work.
>
> > Why does CEC perform worse than vanilla RL?
>
> Vanilla RL and curriculum RL are directly trained on the test distribution for tens of thousands of episodes until convergence. They are further used as source agents to generate offline training data. Therefore, these two RL agents should be considered as **oracle**. This fact is stated at L178 in our submission. By contrast, our CEC agents are only trained for several hundred episodes generated by them. For our task-difficulty-based curriculum, test tasks are even outside the training distribution (Table 1 on page 5). Even so, our CEC agents perform comparably to and even outperform RL oracles, confirming the superior data efficiency achieved by our methods.
>
> > What factors contribute to the subpar performance of Learning progress?
>
> To make the discussion more concrete, we visualize different policies on Irreversible Path. Videos can be found [here](https://tinyurl.com/yypwytt). We find that by keeping a task-difficulty-based curriculum in context, the agent is able to learn critical skills such as visual navigation and long-horizon planning from relatively easier tasks, then apply them to the most challenging setting. This is demonstrated by the fact that the agent can navigate to gradually distant goals. Evaluated on the same difficulty, the agent resulting from a learning-progress-based curriculum sometimes makes progress toward the goal. The relatively inferior performance could be due to the fact that source RL agents, which are used to generate training data, also struggle in this task.
>
> > The definition of learning progress
>
> As mentioned in L77 of our paper, we view all RL tasks in this work as goal-reaching problems. Therefore, learning progress can be defined as the improvement in task success rate or reward over time. We will clarify this in our final version.
>
> > Curriculum pertaining to environmental change
>
> The field of automatic environment design, albeit interesting, is beyond the scope of this work. Since we focus on improving the learning efficiency of Transformer agents on sub-optimal data, our evaluation has become more comprehensive after we added DT and AT baselines, thanks to your suggestion!
>
> Incorporating automatically generated environments could be an exciting future extension of CEC, and we will discuss PAIRED [10] in our final version.
>
> **References**
> - [1] Mirowski et al., Learning to Navigate in Complex Environments, ICLR 2017.
> - [2] Li et al., Hierarchical Reinforcement Learning with Advantage-Based Auxiliary Rewards, NeurIPS 2019.
> - [3] Liu and Abbeel, Emergent Agentic Transformer from Chain of Hindsight Experience, ICML 2023.
> - [4] Chen et al., Decision Transformer: Reinforcement Learning via Sequence Modeling, NeurIPS 2021.
> - [5] Beattie et al., DeepMind Lab, 2016.
> - [6] Mandlekar et al., What Matters in Learning from Offline Human Demonstrations for Robot Manipulation, CoRL 2021.
> - [7] Espeholt et al., IMPALA: Scalable Distributed Deep-RL with Importance Weighted Actor-Learner Architectures, ICML 2018.
> - [8] Shafiullah et al., Behavior Transformers: Cloning k modes with one stone, NeurIPS 2022.
> - [9] Chi et al., Diffusion Policy: Visuomotor Policy Learning via Action Diffusion, RSS 2023.
> - [10] Dennis et al., Emergent Complexity and Zero-shot Transfer via Unsupervised Environment Design, NeurIPS 2020.

---

> > ### Comment · Reviewer_hTvj · 2023-08-16
> > **Acknowledgement of the rebuttal**
> >
> > Thank you for your response, which addressed my concerns. These clarifications should be included in the next version. I will raise my score to 6.

---

> ### Author Response · Authors · 2023-08-15
> **Follow-up on our response**
>
> Dear reviewer,
>
> As the discussion stage is ending soon, we wonder if our response answers your questions and our extra experiments address your concerns? If yes, would you kindly consider raising the score? Thanks again for your very constructive and insightful feedback!

---

### Official Review · Reviewer_xTZL · 2023-07-07

**Soundness:** 2 fair
**Presentation:** 3 good
**Contribution:** 3 good
**Rating:** 5
**Confidence:** 3

**Summary:**

This work aims to study mechanisms of cross-episode attention to effectively learn to improve polices by training on contexts containing gradually improving trajectories.

**Strengths:**

On the whole, this paper is well written.  The topic of transformers in-context learning as an approach to planing and RL is of increasing importance. This work provides another important datapoint in that space for a technique which has scientific value to try.

**Weaknesses:**

The main weakness with this work is its lack of comparison to existing approaches in the space.  The empirical results are only compared against approaches that are not intending to take advantage of in-context learning.  This looses most of the scientific value of the work when several other prominent works have used the in-context learning capabilities of transformers in various ways to great success. It is known that in-context learning can help with these sorts of tasks, the question which this paper is in the position to answer, is if this way of applying in-context learning works better or worse than the others which have been tried. Even if this approach is importantly distinct from those, it is important to compare against the other natural approaches that have already been tried, such as Algorithmic Distillation or AdA, to understand if this approach to applying in-context learning works better or worse than other approaches.

On a similar note, the paper is often vague about how their approach is different from other approaches in the literature. Specifically the distinction between "test time meta-learning" and "generalization across varying test configurations in each episode" is difficult to understand.

**Questions:**

How does your RL approach in the DM labs differ from the general approach from Adaptive Agents Team et al?

**Limitations:**

The paper could include more discussion of the feasibility of getting data with the requisite ordering of poor data first, better data next, best data last.  It could also be useful for the paper to include some mention of the limitations of the imitation paradigm.

---

> ### Author Rebuttal · Authors · 2023-08-09
>
> Thank you for your constructive feedback. We address your concerns in detail below and will update our paper accordingly.
>
> > Lack of comparison to existing approaches
>
> Thank you for pointing this out. We compare against two more relevant baselines, Agentic Transformer (AT [1]) and Decision Transformer (DT [2]). Due to the limit on reply length, please refer to the [global response](https://tinyurl.com/4bc2469z) for comparison and discussions.
>
> > Differences between Algorithm Distillation
>
> Thanks for raising this question. First, we would like to highlight that we have different focus and scope. AD [3] focuses on learning a meta-RL agent that demonstrates in-context improvement during test time without gradient updates. Instead, we focus on how to improve the data efficiency for Transformer-based agents by explicitly formulating data in curricular sequences. To achieve this goal, we introduce and validate three solutions in RL and IL settings, which are underexplored in AD.
>
> Second, we admit that our learning-progress-based curriculum bears a resemblance to AD at first glance, but they still differ in several angles, as detailed below.
>
> - On data generation. AD requires **N** different single-task source agents to generate data, which amounts to N different copies of model weights. This is evident in the second requirement and line 1 in Algorithm 1 in the AD paper. However, our learning-progress-based curriculum makes minimal assumptions about the diversity of source agents. In fact, we only require a single multi-task agent to generate training data. This makes our method more applicable since it requires less storage and time.
>
> - On evaluation. AD is supposed to only work with a single task instance at test time. This is evident in line 12 in Algorithm 1 in the AD paper. In other words, every time the environment is reset, the agent will be spawned at exactly the same initial location. Goal location and other environmental transitions also keep the same for all evaluation episodes. However, our agent is evaluated with new and different task instances every episode, meaning that the agent's initial location, goal location, and other factors change every test episode. Therefore, our test setting is **strictly harder** than AD’s.
>
> Due to the above two differences, we do not think our work is directly comparable to AD. We will still conceptually compare our learning-progress-based curriculum against AD and clarify the discussion in the camera-ready version.
>
> > Differences between Adaptive Agent Team et al. [4]
>
> We would like to highlight the differences between AdA [4] from the following perspectives.
>
> - Scope and focus. Similar to how our work compares against AD, we focus on improving the data efficiency for Transformer-based agents by explicitly formulating data in curricular sequences. We introduce and validate three solutions toward this goal. Our method also works with imitation learning settings — robotic manipulation from human demonstrations — which meta-RL literature such as AdA does not explore.
>
> - Online data generation vs. offline data generation. AdA is trained by online RL (Muesli [5], specifically), where the agent is allowed to learn from experiences through trial-and-error. However, our cross-episodic curriculum method is offline behavior cloning instead, where training data are first generated and then fixed and used for learning.
>
> - Reproducibility and open-sourceness. AdA is developed and evaluated only on XLand [6], which has not been open-sourced and accessible by the community. Their algorithm implementation is also not open-sourced. By contrast, our method is developed and validated with open-sourced environments (DMLab [7] and RoboMimic [8]). We also open-source our algorithm implementation to contribute to the community. It is easy and straightforward for the community to reproduce our results, and further develop follow-up techniques on other domains.
>
> > More discussion of the feasibility of getting curricular data
>
> Thanks for your suggestion. We discuss the feasibility of obtaining each curriculum one by one in the [global response](https://tinyurl.com/4bc2469z). We will move the discussion to the main text in the camera-ready version.
>
> **References**
> - [1] Liu and Abbeel, Emergent Agentic Transformer from Chain of Hindsight Experience, ICML 2023.
> - [2] Chen et al., Decision Transformer: Reinforcement Learning via Sequence Modeling, NeurIPS 2021.
> - [3] Laskin et al., In-context Reinforcement Learning with Algorithm Distillation, ICLR 2022.
> - [4] Adaptive Agent Team, Human-Timescale Adaptation in an Open-Ended Task Space, ICML 2023.
> - [5] Hessel et al., Muesli: Combining Improvements in Policy Optimization, ICML 2022.
> - [6] Open-Ended Learning Team, Open-Ended Learning Leads to Generally Capable Agents, 2021.
> - [7] Beattie et al., DeepMind Lab, 2016.
> - [8] Mandlekar et al., What Matters in Learning from Offline Human Demonstrations for Robot Manipulation, CoRL 2021.

---

> > ### Comment · Reviewer_xTZL · 2023-08-18
> > **Response to Authors**
> >
> > Thank you for your rebuttal. You have partially addressed my concerns on the novelty of this work. However, it is still problematic that this is largely written as if the in-context learning ability of transformers has not been used in RL when there have been many notable successful applications. As such I find it difficult to raise my score above a 5 (borderline accept) without a more narrow and clear framing of the contributions relative to other methods of achieving in-context learning in RL.

---

> ### Author Response · Authors · 2023-08-15
> **Follow-up on our response**
>
> Dear reviewer,
>
> As the discussion stage is ending soon, we wonder if our response answers your questions and our extra experiments address your concerns? If yes, would you kindly consider raising the score? Thanks again for your very constructive and insightful feedback!

---

### Author Rebuttal · Authors · 2023-08-09

# Global Response
We sincerely thank all reviewers for their thoughtful and constructive feedback. We really appreciate that all reviewers find our idea novel and important for Transformer-based agents. We attach updated versions of Figs 3 and 4 in the one-page PDF. In our response to each reviewer below, we address their individual questions and comments. We will update the paper and supplementary PDFs with revisions accordingly. We welcome any follow-up discussions!
## More Baseline Comparison
We extend the main experiments on DMLab with two more baselines, Agentic Transformer (AT [1]) and Decision Transformer (DT [2]). AT is a **concurrent** work with the preprint version released after NeurIPS submission and conference version published at ICML in July. It is closely related to our work by training Transformers on sequences of trajectory ascending sorted according to their rewards. DT is a popular Transformer-based offline RL method that conditions on return-to-go and does not use any forms of cross-episodic curriculum. We control data, model capacity, training strategy, etc. to be the same. We denote AT and DT trained on data consisting of a mixture of task difficulties as “AT (Mixed Difficulty)” and “DT (Mixed Difficulty)”. Note that these data are the same used to train “Ours (Task Difficulty)”. Similarly, we denote AT and DT directly trained on test difficulty as “AT (Single Difficulty)” and “DT (Single Difficulty)”. These data are the same used to train “Ours (Learning Progress)”. Figures are attached in the PDF. Averaged numerical results are presented as below.

*Fig 3 averaged results.*
|  | Ours (Task Difficulty), Auto | Ours (Task Difficulty), Fixed | Ours (Learning Progress) | DT (Mixed Difficulty) | DT (Single Difficulty) | AT (Mixed Difficulty) | AT (Single Difficulty) | BC w/ Expert Data | RL (Oracle) | Curriculum RL (Oracle) |
|:---:|:---:|:---:|:---:|:---:|:---:|:---:|:---:|:---:|:---:|:---:|
| Average | 51.4 | **54.4** | 32.4 | 35.3 | 11.7 | 42.7 | 33.4 | 14.2 | 40.6 | 50.6 |

*Fig 4 averaged results.*
|  | Ours (Task Difficulty) | Ours (Learning Progress) | DT (Mixed Difficulty) | DT (Single Difficulty) | AT (Mixed Difficulty) | AT (Single Difficulty) | BC w/ Expert Data | RL (Oracle) | Curriculum RL (Oracle) |
|:---:|:---:|:---:|:---:|:---:|:---:|:---:|:---:|:---:|:---:|
| Average | **39.6** | 27.78 | 31.8 | 13.6 | 39.4 | 29.2 | 18.1 | 30.0 | 37.6 |

We can see that our method with task-difficulty-based curriculum performs the best during evaluation (Fig 3, the first table above), confirming the benefit over the concurrent AT approach that leverages chain-of-hindsight experiences. When compared to DT, it outperforms by a significant margin, which suggests that our cross-episodic curriculum helps to squeeze learning signals that are useful for downstream decision-making. When evaluating the generalization ability (Fig 4, the second table above), our method performs better than the concurrent AT baseline and achieves significantly better results than other baselines. This empirically suggests that CEC helps to learn policies that are robust to environmental perturbations and can quickly generalize to new changes.
## Policy Visualization
We visualize agents' behavior on the DMLab task Irreversible Path [here](https://tinyurl.com/yypwytt). The agent trained from task-difficulty-based curriculum can navigate to gradually more distant goals. Keeping the curriculum in context helps to distill useful exploration and long-horizon planning skills, which are critical to the success of the most difficult level. Evaluated on the same difficulty, the agent resulting from a learning-progress-based curriculum sometimes makes progress. The relatively inferior performance could be due to the fact that source RL agents, which are used to generate training data, also struggle in this task. DT and AT demonstrate even worse behavior. They usually choose the wrong direction and get stuck forever.
## Discussion on the Feasibility of Obtaining Curricular Data

- Learning-progress-based curriculum. RL agents generally monotonically improve over time, so they naturally generate increasingly better data. Our curriculum is constructed by a series of checkpoints over the course of training. There is no additional assumption to formulate this curriculum.

- Task-difficulty-based curriculum. a) For environments with parameterized difficulty, it is straightforward to create a schedule based on the difficulty parameter to formulate the curriculum as we did in this work. b) For environments where the difficulty is not parameterized, methods such as those in [3] can be used. An exciting avenue of future research would be applying our method to real-world tasks without explicitly defined difficulty.

- Expertise-based curriculum. One limitation is the need to estimate the demonstrators’ proficiency. Some IL benchmarks, such as RoboMimic [4], already have proficiency labels. Despite this, to apply our method more broadly, one way to approximate proficiency is to rank trajectories by completion time. Moreover, in practice, a demonstrator’s proficiency naturally increases when collecting more data: from not being familiar with the teleoperation systems or the task to collecting data with muscle memory [5]. This progress could provide a rich learning signal using CEC.

We will move this discussion to the main text in the final version.

## References
- [1] Liu and Abbeel, Emergent Agentic Transformer from Chain of Hindsight Experience, ICML 2023.
- [2] Chen et al., Decision Transformer: Reinforcement Learning via Sequence Modeling, NeurIPS 2021.
- [3] Kanitscheider et al., Multi-task curriculum learning in a complex, visual, hard-exploration domain: Minecraft, 2021.
- [4] Mandlekar et al., What Matters in Learning from Offline Human Demonstrations for Robot Manipulation, CoRL 2021.
- [5] Mandlekar et al., RoboTurk: A Crowdsourcing Platform for Robotic Skill Learning through Imitation, CoRL 2018.

---

### Decision · Program_Chairs · 2023-09-21

**Decision:**

Accept (poster)

**Comment:**

The paper proposes cross-episode curricular approach to boost learning of transformer-based agents. The work is interesting in its perspective and was made more convincing by adding two extra baselines during the rebuttal, but could be improved by making it clearer under what circumstances CEC is expected to be helpful.